# Immune Evaluation of Avian Influenza Virus HAr Protein Expressed in *Dunaliella salina* in the Mucosa of Chicken

**DOI:** 10.3390/vaccines10091418

**Published:** 2022-08-29

**Authors:** Inkar Castellanos-Huerta, Gabriela Gómez-Verduzco, Guillermo Tellez-Isaias, Guadalupe Ayora-Talavera, Bernardo Bañuelos-Hernández, Víctor Manuel Petrone-García, Isidro Fernández-Siurob, Gilberto Velázquez-Juárez

**Affiliations:** 1Programa de Maestría y Doctorado en Ciencias de la Producción y de la Salud Animal, Facultad de Medicina Veterinaria y Zootecnia, Universidad Nacional Autónoma de México, Ciudad Universitaria, Ciudad de Mexico 04510, Mexico; 2Departamento de Medicina y Zootecnia de Aves, Facultad de Medicina Veterinaria y Zootecnia, Universidad Nacional Autónoma de México, Avenida Universidad 3000, Ciudad de Mexico 04510, Mexico; 3Department of Poultry Science, University of Arkansas, Fayetteville, AK 72701, USA; 4Centro de Investigaciones Regionales Dr. Hideyo Noguchi, Universidad Autonoma de Yucatan (UADY), Merida 97000, Yucatan, Mexico; 5Escuela de Veterinaria, Universidad De La Salle Bajío, Avenida Universidad 602, Lomas del Campestre, Leon 37150, Guanajuato, Mexico; 6Departamento de Ciencias Pecuarias, Facultad de Estudios Superiores Cuautitlán UNAM, Cuautitlan 54714, Mexico; 7Viren SA de CV, Presidente Benito Juárez 110B, José María Arteaga 76135, Queretaro, Mexico; 8Departamento de Química, Centro Universitario de Ciencias Exactas e Ingenierías, Universidad de Guadalajara, Boulevard Marcelino Garcia Barragan #1421, Guadalajara 44430, Jalisco, Mexico

**Keywords:** avian influenza, S-IgA, immunoglobulin, mucosal, hemagglutinin, recombinant protein

## Abstract

Avian influenza (AI) is a serious threat to the poultry industry worldwide. Currently, vaccination efforts are based on inactivated, live attenuated, and recombinant vaccines, where the principal focus is on the type of virus hemagglutinin (HA), and the proposed use of recombinant proteins of AI virus (AIV). The use of antigens produced in microalgae is a novel strategy for the induction of an immune response in the mucosal tissue. The capacity of the immune system in poultry, particularly in mucosa, plays an important role in the defense against pathogens. This system depends on a complex relationship between specialized cells and soluble factors, which confer protection against pathogens. Primary lymphoid organs (PLO), as well as lymphocytic aggregates (LA) such as the Harderian gland (HG) and mucosa-associated lymphoid tissue (MALT), actively participate in a local immune response which is mainly secretory IgA (S-IgA). This study demonstrates the usefulness of subunit antigens for the induction of a local and systemic immune response in poultry via ocular application. These findings suggest that a complex protein such as HAr from AIV (H5N2) can successfully induce increased local production of S-IgA and a specific systemic immune response in chickens.

## 1. Introduction

Avian influenza (AI) is classified from a respiratory to a systemic disease, with a high economic and health impact on the poultry industry worldwide [1]. Clinical presentations of AI range from undetected to serious illness with high mortality depending on the virus subtype and factors such as age, sex, and the commercial purpose of the poultry [2]. The AI virus (AIV), a member of the *Orthomyxoviridae* family [3], presents the following two glycoproteins on its surface: hemagglutinin (HA) and neuraminidase (NA) [4]. HA, the principal surface antigen, and the most abundant protein on the surface of viruses [5], is a trimer (≈225 kDa) made up of identical monomers (≈75 kDa) that forms a trimeric, elongated, rod-shaped protein organized into two polypeptides—HA1 and HA2 polypeptides—and intertwined in a core-helical coiled-coil (stem-like domain) with N-linked oligosaccharide side chains and three globular heads in the monomeric structure of HA0 [6]. The globular head of HA is the viral receptor, which interacts with a cellular receptor (sialic acid) [6,7]. The vaccination strategy against AI is largely based on the use of the HA subtype circulating in the susceptible population [8] through the use of inactivated virus vaccines and recombinant virus vaccines [9,10]. Another strategy proposed is the expression of recombinant antigenic proteins from AI [10]. In particular, the expression of HA in heterologous systems demonstrates the ability of these systems to produce recombinant proteins as possible antigens against AI with different approaches [6,8,11,12,13,14] including a heterologous protein expression system with high potential in various applications such as microalgae [15]. Traditional systems present particularly undesirable characteristics including high production costs, undesirable post-translational modifications, prolonged cultivation times, and the lack of glycosylation in proteins in some cases [16]; however, microalgae present several advantages for the expression of proteins with particular characteristics, such as the HA protein [17]. 

The immune system of poultry, especially in the mucosal tissue, plays a very important role in the defense against pathogens, whose entry or presence in the mucosa is a constant problem in animals [18]. In general, the immune system depends on a complex relationship between specialized cells and soluble factors, working together to confer protection against pathogens [1,19,20]. In particular, the mucosal immune system of birds has similarities with mammals, such as the presence of mucosa-associated lymphoid tissue (MALT), with some exceptions [19]. Among these, we can highlight the presence of primary lymphoid organs, such as the bursa of Fabricius (FB) and lymphocytic aggregates such as the Harderian gland (HG). The HG is observed in all vertebrates with the exception of fishes, some amphibians and primates, whereas in poultry HG is the major contributor to ocular Ig production [20]. HG is a dominant orbital gland in birds, with a strap-like structure, located anatomically ventral and posteromedial to the eyeball, and connected to the medial angle of the nictitating membrane [21]. HG is responsible for the production of secretory IgA (S-IgA) [22,23], mainly as a result of local stimulation [24]. Therefore, HG participates in the immune response in mucous membranes, mostly in the eyes and upper respiratory tract [24]. S-IgA, a polypeptide of two monomers IgA, is the predominant Ig isotype on most mucosal surfaces; due to its molecular stability and strong anti-inflammatory properties, S-IgA is an ideal protective immunity component of mucosal surfaces [25]. S-IgA and immunoglobulin M (IgM) represent an important part of the defenses at the mucosal level due to their function of preventing the adhesion of pathogens such as bacteria and viruses, limiting their pathogenic effect, and neutralizing and facilitating their excretion [26]. However, the parenteral administration of antigens in many cases is inefficient for its induction, hence the mucosal immune system is considered as separate from the systemic immune system [27]. The administration of antigens via mucosa led to the stimulation of a systemic response [28]; however, the main focus is the local stimulation of an immune response due to the need for protection in these tissues. 

In the case of recombinant antigens from microalgae, due to all advantages over other antigen-production systems [29], it is possible to explore their use as mucosal vaccines, as a complementary stimulation pathway for parenteral immunization [30]. This study aimed to examine the local and systemic immune response of the recombinant HAr protein of AIV expressed in microalgae (*Dunaliella salina*) in chickens via mucosal administration.

## 2. Materials and Methods

### 2.1. Expression of HAr Recombinant Protein 

HAr expression was performed in a previously reported study. Briefly, a sequence of the HA gene of the reference strain A/chicken/Hidalgo/28159-232/1994 (H5N2) Genbank # CY006040.1) was used for the agroinfiltration protocol for expression in *D. salina* according to the method of Castellanos-Huerta et al. [15]. 

### 2.2. Preparation of Antigens for Mucosa Application 

#### 2.2.1. Preparation of Virus-Inactivated Antigens

Virus-inactivated antigens for mucosa application were prepared as follows: allantoic fluid from 13-day-old embryos was inoculated with AIV H5N2 A/chicken/Hidalgo/28159-232/1994 (H5N2), and low pathogenicity AIV (LPAIV) with a minimum viral titer of 10^9.1^ 50% chicken embryo infectious dose (CEID)/1 mL was harvested and inactivated with 0.2% formaldehyde for 1 h under constant stirring at room temperature and stored for 24 h at 4 °C, with a minimum titer of hemagglutination test of 512 hemagglutination units (HAU)/mL, then it was mixed with the immunomodulator A1 at a ratio of 98:2 (*v*/*v*, allantoic fluid: A-1), according to the method of Fernandez-Siurob et al. [31]. Immunomodulator A1 was prepared from the lipopolysaccharide mixture as described by Westphal et al. [32], which was extracted from *Escherichia coli* strains characterized on an outbreak field [33].

#### 2.2.2. Preparation of HAr Antigen Produced in *Dunaliella salina*

The production of microalgae biomass was carried out in 15 L of PKS (phosphate–potassium–sodium) modified medium in flasks with magnetic agitation (Corning PYREX glass Proculture^®^ Spinner Flasks 15,000 mL. (New York City, NY, USA). The biomass from the culture was harvested by centrifugation at 13,000 rpm for 15 min at 4 °C and stored at −80 °C until ready for use. For antigen HAr’s preparation protocol, biomass was ground using a mortar and pestle in liquid nitrogen, suspended in lysis buffer (1% SDS, 10 mM Tris-MOPS, 2 mM MgCl2, 10 mM KCl pH 7.5, and 2 mM PMSF, added before use), and treated to obtain total protein soluble (TPS) according to the methods of a previous report [15]. According to the quantitative densitometry of proteins stained with Coomassie blue, approximately 1.277 mg of recombinant protein was recovered from 10 g wet weight (WW) of *D. salina*. TPS from *D. salina* culture extracts was filtered through a PVDF membrane pore diameter of 0.22 μm and verified by centrifugation. Recombinant protein HAr was concentrated by performing ultra-filtering; upon the 10-fold increase in the sample a sufficient concentration was achieved. The samples were analyzed using a Quick Start™ Bradford Protein assay (Bio-Rad, Hercules, CA, USA) mixed with the designated immunomodulator A1, and homogenized to a final concentration of ~25 μg of recombinant protein HAr, each in 50 μL. The antigen and immunomodulator A1 were combined at a ratio of 98:2 (*v*/*v*, recombinant protein: A1), mixed for 10 min, and maintained at 4°C until use. One dose consisted of 100 μL of the antigen preparation (HAr), with 50 µL in each eye. A total of 50 μg of recombinant protein HAr was considered as a dose per animal, as previously observed [11]. 

### 2.3. Animal Experiment 

The trial was performed to determine the antigenicity of the recombinant antigen HAr compared with a virus-inactivated antigen on mucosal application, as previously reported [31]. The animal experimentation consisted of 60 laying hen chickens divided into groups of 20 for treatments A, B, and C. In all of the experiments, animals were housed in poultry coops and divided according to the applied treatments. Treatment A consisted of the positive control via the ocular route with 100 μL (50 µL in each eye) of virus-inactivated antigen/adjuvant prepared as described above. Treatment B consisted of the application of 100 μL (total of 50 μg of recombinant protein HAr) protein/adjuvant (50 µL in each eye) via the ocular route. Treatment C consisted of the negative control including a treatment of PBS/A1 combined at a ratio of 98:2 (*v*/*v*, PBS/A1), mixed for 10 min, and maintained at 4 °C until its use via the ocular route (total of 100 μL of PBS, 50 µL in each eye). For this experiment, a calibrated dropper was used to deposit a 50 µL drop in each eye according to the manufacturer’s instructions. In all of the experiments, environmental and animal management conditions were upheld according to the standards of laying chicken breeding. Feed and water were supplied ad libitum throughout the entire experiment. For the trial, treatment A (virus-inactivated antigen), treatment B (HAr), and treatment C (PBS/A1) were applied at 14 and 21 days of age. For each treatment, 10 birds were sampled for total S-IgA quantitation in lachrymal fluid at 7 and 14 days post-treatment (DPT) (28 and 36 days of age); the samples were analyzed by performing an ELISA test according to the method of Merino-Guzman et al. [34]. A total of 12 birds were bled for serum at 14 and 21 DPT (36 and 42 days of age) for the inhibition of the hemagglutination (HI) assay to determine the serum immunoglobulin levels against AIV H5N2 [35]. All samples were stored at −80 °C until use. During the experiment, the experimental animals were handled according to ethical protocols for animal welfare [36].

### 2.4. ELISA Test of S-IgA from Lachrymal Fluid

Quantification of total S-IgA in lachrymal fluid samples was carried out using a commercial ELISA kit (Catalog #: E33-103, Bethyl Laboratories, Inc., Montgomery, TX, USA) following the manufacturer’s guidelines. Samples were diluted at 1:1000 (*v*/*v* lachrymal fluid: dilution buffer), prior to their use in the ELISA test. Concentrations of S-IgA were expressed as nanograms of total S-IgA per ml (ng/mL).

### 2.5. Inhibition of Hemagglutination (HI) Assay 

The stimulation of the systemic immune response of treatments A, B, and C was evaluated using the HI assay expressed as the geometric mean titer (GMT). All measurements of serum AI-specific antibody levels were analyzed with 4 HAU of the AIV A/chicken/Hidalgo/28159-232/1994 (H5N2) strain, according to the WHO Manual on Avian Influenza Diagnosis and Surveillance [37]. 

### 2.6. Statistical Analyses

Absorbance results in the ELISA test (for total S-IgA in the lachrymal fluid) were extrapolated onto a polynomial calibration chart according to the supplier’s instructions. The data confirmed normal distribution (Shapiro–Wilk test) and homoscedasticity (Levene test). Consequently, the data were subjected to a parametric test and one-way ANOVA, followed by Tukey’s multiple comparison test with the level of statistical significance set at *p* < 0.05.

## 3. Results

### 3.1. ELISA Test of S-IgA from Lachrymal Fluid

The quantification of total S-IgA determined by performing the ELISA test at 7 DPT in the lachrymal fluid of animals on treatments A (48,334 ng/mL ± 4196 ng/mL) and B (44,759 ng/mL ± 8002 ng/mL), showed a statistical difference (*p* > 0.05) compared with treatment C (24,224 ng/mL ± 3496 ng/mL) (Figure 1A). Samples obtained at 14 DPT showed an increase in the local immune response for the concentration of total S-IgA in treatment A (57,486 ng/mL ± 7192 ng/mL), with a statistical difference (*p* > 0.05) compared with treatment B (43,043 ng/mL ± 5733 ng/mL) and treatment C (35,633 ng/mL± 6195 ng/mL) (Figure 1B). No statistical difference was observed between treatments B and C. 

### 3.2. HI Assay from Serum 

According to the results, for treatment A (GMT = 8.97 ± 1.30), treatment B (GMT = 12.70 ± 1.98), and treatment C (GMT = 10.67 ± 1.59) at 14 DPT no statistical difference was observed (*p* > 0.05) (Figure 2A). The systemic immune response observed at 21 DPT presents a statistical difference (*p* > 0.05) by comparing treatment A (GMT = 38.05 ± 1.54), treatment B (GMT = 33.90 ± 1.86), and treatment C (GMT = 6.72 ± 1.54) (Figure 2B).

## 4. Discussion

A description of the response of the mucosal immune system is necessary for the understanding of several pathologies of infectious origin, particularly in the case of AI [38,39]. As is known, the main route of entry for pathogens into an organism is via the mucosa tissue [40]. Therefore, this tissue represents the first barrier that needs to be overcome in order to achieve infection and the generation of infectious diseases. For this reason, the mucosa requires a highly specialized immune system for the correct response and protection against various pathogens [38,39]. In the case of birds, these lymphoid organs present some variations with respect to other species [38,41]; however, the specialized tissue plays a vital role in homeostasis, as in other species. The ocular route for vaccination is widely used in the poultry industry as an option for diseases such as Newcastle disease and avian bronchitis [23,24] due to its practicality, as well as the relevance and participation of lymphocytic aggregates, e.g., HG [20,22,23] The active participation of HG in the production of an adaptive immune response after ocular administration represents a tool for the local application of antigens [11,42] due to the relevance in the participation of S-IgA in the protection against infections, as well as the induction of immunity after vaccination [43]. Nevertheless, the measurement of this Ig subtype in lachrymal fluid samples [34] is not a regular practice.

The main objective of this study was to determine the antigenicity of a recombinant protein compared with native viral antigens of AIV. The recombinant protein (HAr) was previously produced in a protein-recombinant expression model [15] with advantages in terms of production costs, expression levels, as well as its possible use in the oral vaccination of animals [29]. Therefore, mucosal vaccination was demonstrated in an animal model (poultry) due to the importance of local mucosal immunity against diseases caused by respiratory viruses such as AI [44]. 

The total S-IgA ELISA kit was utilized to determine the reactivity of the ocular application of antigens by stimulating eye-associated lymphoid tissue [24]. Despite the non-specific detection of S-IgA against AI, it was possible to determine different levels of S-IgA stimulation during the application of both viral antigens and the recombinant protein. At 7 DPT, an increase in the levels of total S-IgA in the lachrymal fluid samples was detected following the application of the viral protein and recombinant protein, compared to the negative control group, thereby demonstrating HG stimulation in the presence of the administered antigens [45]. For the second sampling at 14 DPT, an antigen of viral origin showed a significant increase compared to the recombinant antigen and the negative control group in the total S-IgA levels, due to antigenic components of AIV (proteins NA, M) as well as internal viral proteins [46]. The total S-IgA values induced at 14 DPT by the recombinant HAr protein maintained the same level compared to 7 DPT sampling; however, bacterial components in the immunomodulator used called A1 were able to stimulate an adaptive immune response against these bacterial components [47], detectable by the total S-IgA ELISA kit in the negative control group. The results of total S-IgA at 7 DPT showed the ability of viral proteins and the HAr protein to stimulate local immunity. Nevertheless, at 14 DPT, due to the use of an adjuvant based on bacterial components and the absence of viral components, the results observed reflect a local immune response with less activity from the HAr protein compared to the complete viral antigen [46], thus reflecting the necessity to improve HAr subunit antigens, including the determination of SIgA-specific antigens against AIV antigens for an accurate evaluation.

The HI assay was proposed to determine the ability to stimulate a systemic immune response through ocular administration of viral proteins and a recombinant protein HAr. [11]. The results observed at immunization at 14 days DPT with viral proteins, recombinant protein and the application of PBS/A1, showed no significant difference. However, at 21 days DPT, an increase in HI titers was observed in animals subjected to treatments with the virus protein and recombinant HAr protein, compared to the negative control group, demonstrating that immune stimulation via the ocular route is relevant for the local immune response [48]. This was due to the presence of a lymphocyte aggregate HG, [38] as well as its participation in the presentation of antigens to the immune system for the generation of the humoral response at the systemic level [49]. As observed in the measurement of total S-IgA in lachrymal fluid, the stimulus-induced generation of a systemic immune response was also influenced by the participation of other viral antigens present in the sample of AI [50]; therefore, the presence of a single subunit antigen presents an antigenic capacity not similar to a complete viral native antigen. 

The increase in GMT titers in the HI assay upon the application of recombinant antigens demonstrates AI-specific systemic immune stimulation by the ocular administration of a recombinant protein [49] since the HI assay measures the reactivity of stimulated antibodies to AI-specific immunization [37]. Due to the immunological performance of HAr compared to viral proteins administered under the same adjuvant and application route, it is necessary to continue its study as a possible recombinant antigen for AI before considering viral challenge experiments in animals. The titers obtained from the HI assay, induced by the application of viral antigens and recombinant protein HAr, are suggestive of the minimum protection titers (GMT ≥ 32) [51]. However, various experiments are required to determine the minimum protection level.

## 5. Conclusions

The present study described the antigenic capacity of the recombinant protein HAr, applied by the ocular route in experimental animals, in comparison with a complete viral antigen. The results indicate the capacity of protein HAr for stimulation of the local and systemic immune system in an ocular application, for its recognition as a viral protein by HG, as well as its processing and induction of an antibody-specific immune response. The antibodies generated at the serum level proved to be reactive with a wild-type viral antigen, used in the HI assay, thus demonstrating that protein HAr presents characteristics of an antigen of AI. Nevertheless, viral antigens present more antigenic activity according to the ELISA test S-IgA results and HI assay; therefore, further studies are required to develop a recombinant antigen capable of inducing an immune response similar to that observed in antigens of viral origin. 

The results of total measured S-IgA in this experiment demonstrate the efficiency of a recombinant protein to achieve antigenic stimulation in experimental animals; however, these results require a more specific evaluation, because total quantification is not able to determine the specific stimulation against protein HAr and its relationship with viral antigens, describing no specific local immune response. Therefore, its evaluation requires further study. 

In this experiment, the AIV H5N2 A/chicken/Hidalgo/28159-232/1994 (H5N2) strain complied with the stipulations of the Mexican authority for use in the production of vaccines against AI and experimentation in vitro. Nevertheless, the absence of evidence of viral challenges in experimental animals leaves the capacity of these types of antigens unevaluated in the presence of pathogenic viruses, so their efficacy as vaccine antigens requires permission from the local authority to carry out their evaluation in viral challenges in order to understand its full scope as a possible vaccine antigen.

## Figures and Tables

**Figure 1 vaccines-10-01418-f001:**
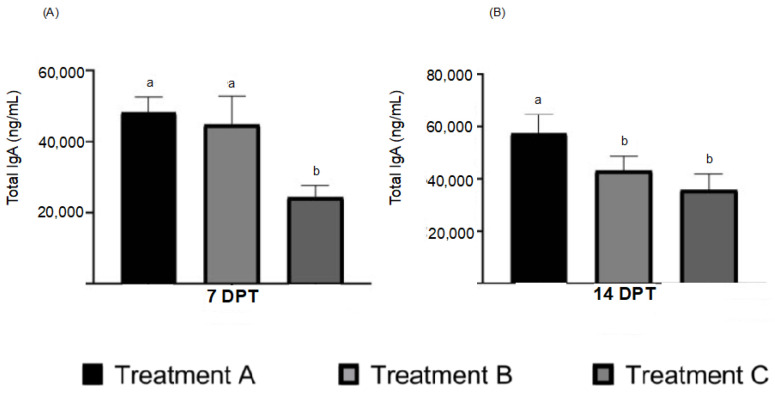
ELISA test of total S-IgA from lachrymal fluid of treatments A, B, and C. The concentration of total S-IgA from lachrymal fluid samples was determined using a commercial ELISA kit. (**A**). ELISA test results of total S-IgA in the lacrimal fluid were observed at 7 days post-treatment (DPT) on treatment A (virus-inactivated antigen), treatment B (HAr), and treatment C (PBS/A1). The results at 7 DPT showed a statistical difference (*p* < 0.05) in total S-IgA concentration in treatment A (48,334 ng/mL ± 4196 ng/mL) and B (44,759 ng/mL ± 8002 ng/mL) compared to treatment C (24,224 ng/mL ± 3496 ng/mL). (**B**). ELISA test results at 14 DPT showed an increase in concentration of S-IgA on treatment A (57,486 ng/mL ± 7192 ng/mL) with a statistical difference (*p* < 0.05) compared with treatment B (43,043 ng/mL ± 5733 ng/mL) and treatment C (35,633 ng/mL ± 6195 ng/mL). No statistical difference was observed in the cases of treatments B and C at 14 DPT. Literals indicate a statistical difference (*p* < 0.05).

**Figure 2 vaccines-10-01418-f002:**
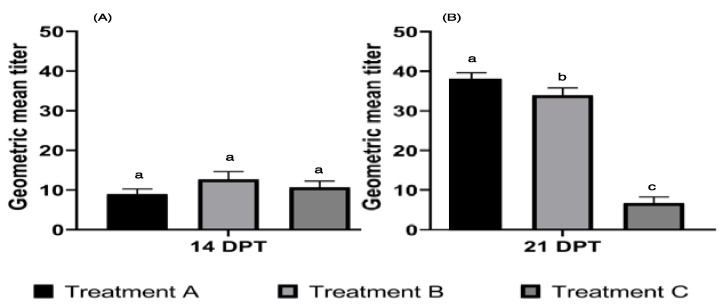
Inhibition of hemagglutination assay (HI) from serum of treatments A, B, and C. Systemic immune response of treatment A (virus-inactivated antigen), treatment B (HAr), and treatment C (PBS/A1) via ocular application at 14 and 21 days post-treatment (DPT) were evaluated by the HI assay expressed in geometric mean titer (GMT). (**A**). Treatment A (GMT = 8.97 ± 1.30), treatment B (GMT = 12.70 ± 1.98), and treatment C (GMT = 10.67 ± 1.59) indicated that no statistical difference was observed (*p* > 0.05) at 14 DPT. (**B**). Immune response at 21 DPT was observed in treatment A (GMT = 38.05 ± 1.54), treatment B (GMT = 33.90 ± 1.86) and treatment C (GMT = 6.72 ± 1.54) with a statistical difference (*p* > 0.05) in comparison to each other. Literals indicate a statistical difference (*p*-value < 0.05).

## Data Availability

Not applicable.

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
