# Peer review of "Immune Evaluation of Avian Influenza Virus HAr Protein Expressed in Dunaliella salina in the Mucosa of Chicken"

_vaccines, 2022, doi:10.3390/vaccines10091418_

Round 1

Reviewer 1 Report

The production of recombinant viral protein by algae, Dunaliella salina, is of high importance, since the algae can be grown in simple manner. The extraction of the recombinant viral protein is more difficult. Application of this immunogenic protein by eye drop is interesting but of low practical importance. Nevertheless, showing that the influenza HA protein can be produce in algae, and this protein can provoke immune response is of importance. The introduction is too long and should be shortened, keep only the relevant information (no ned to the part starting on raw 66 and ends at 75 describing the immune system cells) 

Author Response

We appreciate your comments and suggestions, which are accepted.

Reviewer 2 Report

Castellano-Huerta et al. report on the immune response in chickens following administration of recombinant HAr protein from an H5N2 AIV produced in Dunaliella salina. The authors compare the immune responses of HAr recombinant protein against a formalin inactivated H5N2 against and a negative control all given by intraocular administration. The authors report that both administrations increase total IgA levels at 7 days, but only formalin inactivated AIV gives an increase by 14 days post treatment. An increase in specific serum AIV antibodies, tested via HAI, was observed for both treatments at 21 days post treatment, with a greater increase observed in the inactivated virus group. The report shows the potential utility of Dunaliella salina produced recombinant HAr as a vaccine, however there are some concerns around the methodology and conclusions of the study.

Major Comments:

1.       Only haemagglutinin inhibition was used to quantify the immune response to treatments. This provides no indication on protection from infection. The authors should discuss correlates for protection in the discussion and conclusions. Ideally a challenge study should be used to investigate if the HAr strategy elicits a protective immunity. Other techniques, such as virus neutralisation assays, must be used to investigate the protective nature of antibodies elicited by the three treatments.

2.       Specific IgA antibodies were not investigated, only total IgA, quantifying influenza specific IgA antibodies should be attempted to allow appropriate conclusions to be drawn. The current conclusions are overstated based on presented data.

3.       The advantage of using HAr over inactivated virus is unclear considering worse antibody responses were observed for HAr treatment.

4.       Only one old historic vaccine strain was used for analysis, the rationale for using this strain should be discussed and other strains should be considered to confirm this approach is applicable to contemporary AIVs.

5.       There does not appear to be an ethical statement regarding the animal experimentation.

Minor comments:

1.       Numerous typos throughout, some of the phrasing is not accurate. The manuscript needs to be heavily edited.

2.       The figures need to be consistent in presentations.

3.       The use of AI in reference to the disease, and AIV in reference to the virus is often used interchangeably, this needs correcting

4.       Figure 1. The graphs should be labelled as a multiple panel, with A. and B. indicators.

5.       Both figures would benefit from being displayed with individual values with mean and error bar indicators to better appreciate the spread of the data.

6.       The limit of detections need to be added to both figures.

Author Response

The production of recombinant viral protein by algae, Dunaliella salina, is of high importance, since the algae can be grown in simple manner. The extraction of the recombinant viral protein is more difficult. Application of this immunogenic protein by eye drop is interesting but of low practical importance. Nevertheless, showing that the influenza HA protein can be produce in algae, and this protein can provoke immune response is of importance. The introduction is too long and should be shortened, keep only the relevant information (no ned to the part starting on raw 66 and ends at 75 describing the immune system cells)

Dear Reviewer, #1, thank you very much for the time you have spent reviewing our manuscript. Your

comments are very valuable and helpful for revising our paper and guiding our research. We have studied

those comments carefully and have made corrections, which we hope to meet with approval. The revised

portion in the new version was included and is highlighted in yellow in the reviewed manuscript. The

following is our point-by-point response to reviewers’ comments:

Castellano-Huerta et al. report on the immune response in chickens following administration of recombinant HAr protein from an H5N2 AIV produced in Dunaliella salina. The authors compare the immune responses of HAr recombinant protein against a formalin inactivated H5N2 against and a negative control all given by intraocular administration. The authors report that both administrations increase total IgA levels at 7 days, but only formalin inactivated AIV gives an increase by 14 days post treatment. An increase in specific serum AIV antibodies, tested via HAI, was observed for both treatments at 21 days post treatment, with a greater increase observed in the inactivated virus group. The report shows the potential utility of Dunaliella salina produced recombinant HAr as a vaccine, however there are some concerns around the methodology and conclusions of the study.

Major Comments:

1. Only haemagglutinin inhibition was used to quantify the immune response to treatments. This provides no indication of protection from infection. The authors should discuss correlates for protection in the discussion and conclusions. Ideally, a challenge study should be used to investigate if the HAr strategy elicits protective immunity. Other techniques, such as virus neutralization assays, must be used to investigate the protective nature of antibodies elicited by the three treatments.

This report provides complementary scientific information regarding a recombinant protein expressed in a

microalgae model; These findings allow us to continue with the next stage of experimentation focused on

the evaluation of this type of induced immunity against viral challenges in animal models, which is in the

development phase, where both chick embryo models and experimental animals are considered. We will

seek to obtain information on this degree of protection against viral challenges; It is important to mention

that in Mexico, these tests are carried out under the guidelines of the Mexican government, so their use is

restricted to models that meet previous requirements such as scientific tests that demonstrate their

antigenic reactivity and their safety when used in animals.

We appreciate your comments and suggestions, which are accepted.

2. Specific IgA antibodies were not investigated, only total IgA, quantifying influenza specific IgA antibodies should be attempted to allow appropriate conclusions to be drawn. The current conclusions are overstated based on presented data.

We appreciate your comments and suggestions, which are accepted.

3. The advantage of using HAr over inactivated virus is unclear considering worse antibody responses were observed for HAr treatment.

The research was proposed to compare a viral antigen obtained by a biological model such as chicken-

embryo comparison with a unicellular microalgae model; considering that both models, present clear

differences in cost and management, as well as the risk of contamination, among others. The first results

of its evaluation, even though it performs less efficiently than a wild-type antigen, present the first indications

of the level of efficiency of an alternative model for the production of complex antigens, so its possible use

is the focus of further investigations. We appreciate your comments and suggestions

4. Only one old historic vaccine strain was used for analysis, the rationale for using this strain should be discussed and other strains should be considered to confirm this approach is applicable to contemporary AIVs.

The principal approach of the experiment was to demonstrate the antigenic capacity of a protein obtained

by an alternate and novel method, with practical production advantages, in addition, this first experiment

used an antigen authorized by the Mexican authority for use as vaccines; the use of unauthorized strains

is restricted to other viral types.

5. There does not appear to be an ethical statement regarding the animal experimentation.

We appreciate your comments and suggestions, which are accepted and included.

Minor comments:

1. Numerous typos throughout, some of the phrasing is not accurate. The manuscript needs to be heavily edited.

2. The figures need to be consistent in presentations.

3. The use of AI in reference to the disease, and AIV in reference to the virus is often used interchangeably, this needs correcting

4. Figure 1. The graphs should be labelled as a multiple panel, with A. and B. indicators.

5. Both figures would benefit from being displayed with individual values with mean and error bar indicators to better appreciate the spread of the data.

6. The limit of detections need to be added to both figures.

We appreciate your comments and suggestions, which are accepted and included.

Reviewer 3 Report

Dear editorial board of Vaccines

I hope all of you are fine. Regarding the revision of the Manuscript (vaccines-1831316), titled “Immune Evaluation of Avian Influenza Virus HAr protein expressed in Dunaliella salina in the mucosa of chicken”. Some comments should be replied.

1-    What kind of immunomodulator used during preparation of antigens in materials and methods (line 111). You mentioned in line 255 that it is the fraction of the adjuvant of bacterial origin (A-1); however, this still not clear.

2-    Line 120: the full name of TPS should be written.

3-    Line 164: Please indicate the age (days post vaccination) of the collected serum sampling for HI testing.

4-    How can you explain that the results of SIgA concentrations observed in treatment A (virus-inactivated antigen) as 57486 ng/mL ± 7192 ng /mL, was higher than treatment B (recombinant H5Ds) as 43043 ng /mL± 5733 ng /mL at 14 days post vaccination (Fig. 1, lines 188-190).

5-    The results of HI testing at 14 days post vaccination: Why the HI titers were higher in treatment C: 10.67±1.59 (control negative) than treatment A: 8.97±1.30.

6-    Also, why the HI titers at 21 days were higher in treatment A as 38.05±1.54, than treatments B as 33.90±1.86 compared to 6.72±1.54 for treatment C, although treatment A had only the virus-inactivated antigen and treatment B contained the HA antigen plus adjuvant. 

7-    Vaccination-challenge study should be applied for proper differentiation in protection between the subunit antigens (recombinant H5Ds or HAr protein) and the complete vIA.

8-    Lines 113-134: should be separated under the title of Production of Dunaliella salina microalgae indicating the source of Dunaliella salina.

Author Response

Dear editorial board of Vaccines

I hope all of you are fine. Regarding the revision of the Manuscript (vaccines-1831316), titled “Immune Evaluation of Avian Influenza Virus HAr protein expressed in Dunaliella salina in the mucosa of chicken”. Some comments should be replied.

What kind of immunomodulator used during preparation of antigens in materials and methods (line 111). You mentioned in line 255 that it is the fraction of the adjuvant of bacterial origin (A-1); however, this still not clear.

Dear Reviewer, #2, thank you very much for the time you have spent reviewing our manuscript. Your comments are very valuable and helpful for revising our paper and guiding our research. We have studied those comments carefully and have made corrections, which we hope to meet with approval. Revised portion in the new version was included and is highlighted in yellow in the reviewed manuscript. The following is our point-by-point response to reviewers’ comments:

Line 120: the full name of TPS should be written.

We appreciate your comments and suggestions, which are accepted and included.

Line 164: Please indicate the age (days post-vaccination) of the collected serum sampling for HI testing.

 We appreciate your comments and suggestions, which are accepted and included.

How can you explain that the results of SIgA concentrations observed in treatment A (virus-inactivated antigen) as 57486 ng/mL ± 7192 ng /mL, was higher than treatment B (recombinant H5Ds) as 43043 ng /mL± 5733 ng /mL at 14 days post vaccination (Fig. 1, lines 188-190).

We appreciate your comments. The presence of a complex antigen (virions), makes a virus a better immunogen, due to the presence of several viral proteins, as well as other components (viral genetic material) that participate in the stimulation of the cells present in the associated lymphoid tissues, in addition to the structure and viral conformation, which its better performance as an antigen. The results observed up to this point indicate the capacity of a recombinant subunit antigen, however, its study requires an improvement in the structural design of HAr for its use as a possible vaccine antigen in the future. This research seeks, as a first step, to demonstrate the ability to express complex proteins such as hemagglutinins, in order to improve these antigens in the future with the design of complex antigens such as virus-like particles, to seek a stimulation similar to that observed in complete viruses.

The results of HI testing at 14 days post-vaccination: Why the HI titers were higher in treatment C: 10.67±1.59 (control negative) than treatment A: 8.97±1.30.

We appreciate your comments. Prior to this, it is important to mention that the results of the HI test at 14 days post-vaccination do not present a statistical difference in their analysis, in addition to the fact that at 14 days post-vaccination the production of antibodies is not related to the application of antigens, for the time required to be able to observe a response by Ig's in serum; however, it is interesting to observe that the animals vaccinated with complete viral antigens present slightly low HI levels compared to the negative control group as mentioned, which may be related to the antibody maternal titers of the animals, the inherent difference between them or to the possible reduction of antibodies in the presence of a viral antigen, however since there is no statistical difference, it is not possible to assume this conclusion without further evaluation.

Also, why the HI titers at 21 days were higher in treatment A as 38.05±1.54, than treatments B as 33.90±1.86 compared to 6.72±1.54 for treatment C, although treatment A had only the virus-inactivated antigen and treatment B contained the HA antigen plus adjuvant. 

We appreciate your comments. As previously mentioned, a complex antigen such as the virion has various components, that allow greater antigenic stimulation, compared to a subunit antigen such as the HAr protein, using in both cases the same adjuvant scheme; so it is possible to consider that a subunit antigen, in this case, is not efficient enough, as proteins from viral origin administrated in treatment A, so its study requires improving these production conditions of antigens to overcome the current conditions of this expression model of protein.

Vaccination-challenge study should be applied for proper differentiation in protection between the subunit antigens (recombinant H5Ds or HAr protein) and the complete vIA.

We appreciate your comments, This report provides complementary scientific information regarding a recombinant protein expressed in a microalgae model; These findings allow us to continue with the next stage of experimentation focused on the evaluation of this type of induced immunity against viral challenges in animal models, which is in the development phase, where both chick embryo models and experimental animals are considered. We will seek to obtain information on this degree of protection against viral challenges; It is important to mention that in Mexico, these tests are carried out under the guidelines of the Mexican government, so their use is restricted to models that meet previous requirements such as scientific tests that demonstrate their antigenic reactivity and their safety when used in animals.

Lines 113-134: should be separated under the title of Production of Dunaliella salina microalgae indicating the source of Dunaliella salina.

We appreciate your comments and suggestions, which are accepted and included.

Reviewer 4 Report

This manuscript described an evaluation of the immunogenicity of a recombinant hemagglutinin (HA) from avian influenza virus H5N2. The recombinant protein was synthesized in a transformed single-cell algae species, D. salina. The recombinant HA was inoculated intraocularly into chickens (n=20), and serum samples collected at 7, 14, and 21 days after inoculation. Inactivated virus was used as positive control, and PBS as negative control. The authors claimed that the total secretory IgA (collected from lacrimal fluid) was elevated after intraocular inoculation of native (inactivated virus) or recombinant antigen after 7 and 14 days. Serum hemagglutination-inhibition antibody titers were also elevated at day 14 and 21.

Fatal flaws for this manuscript include incompleteness of data, unreliability of data, and misinterpretation of data:

1) Whereas Fig. 1 showed the lacrimal total IgA level, Fig. 2 showed HI titer, and there were no correlations between these two parameters.

2) The specificity of the sIgA was not determined, i.e., are they HA-specific?

3) The data points were not aligned. Fig. 1 showed day 7 and day 14, however, Fig. 2 showed day 14 and day 21.

4) There was no challenge experiment data, hence a more appropriate evaluation for the immunogenicity cannot be made.

5) For Fig. 1, there was no significant difference between the total sIgA of treatment B to control at day 14. Likewise, the GMT of HI titers were below detection (<10) at day 14 for Fig. 2. 

6) The data at day 21 (Fig. 2) was different from their previous publication (Avian Dis 60: 784, 2016), i.e., the GMT HI titers after immunization (inactivated virus or recombinant HA) were below 20 at day 21 (Fig. 3 in that publication) but were at 40 in the current manuscript.

In summary, this manuscript is incomplete. There was no new scientific knowledge gained. It should be rejected. Of note, these researchers were using a different species of single-cell algae (C. reinhardtii) previously. The reasons for the switch remain unknown. The reasons and possible differences should be fully explained, rather than just reporting similar data, a form of "self-plagiarism".

Author Response

This manuscript described an evaluation of the immunogenicity of a recombinant hemagglutinin (HA) from avian influenza virus H5N2. The recombinant protein was synthesized in a transformed single-cell algae species, D. salina. The recombinant HA was inoculated intraocularly into chickens (n=20), and serum samples collected at 7, 14, and 21 days after inoculation. Inactivated virus was used as positive control, and PBS as negative control. The authors claimed that the total secretory IgA (collected from lacrimal fluid) was elevated after intraocular inoculation of native (inactivated virus) or recombinant antigen after 7 and 14 days. Serum hemagglutination-inhibition antibody titers were also elevated at day 14 and 21.

Dear Reviewer, #3, thank you very much for the time you have spent reviewing our manuscript. Your comments are very valuable and helpful for revising our paper and guiding our research. We have studied those comments carefully and have made corrections, which we hope to meet with approval. The revised portion in the new version was included and is highlighted in yellow in the reviewed manuscript. The following is our point-by-point response to reviewers’ comments:

Fatal flaws for this manuscript include incompleteness of data, unreliability of data, and misinterpretation of data:

Whereas Fig. 1 showed the lacrimal total IgA level, Fig. 2 showed HI titer, and there were no correlations between these two parameters.

We appreciate your comments. The quantification of total S-IgA by ELISA test was proposed for the determination of a local stimulus to the application of a recombinant antigen. Due to the behavior in the induction of S-IgA antibodies to its stimulation, the sampling was considered to start during the first 7 days to 14 days, however, the stimulation of a serological response can be seen from 14- or 21-days post-inoculation. Therefore, the sampling of both serum and local stimuli agrees with what is expected in each system, considering systems with separate immune responses.

The specificity of the sIgA was not determined, i.e., are they HA-specific?

The determination of the SIgA response, in this first trial, was proposed for the detection of this local stimulus and to make an estimate of this stimulation, by means of a standardized commercial kit for S-IgA of poultry. The results of this evaluation show that this stimulus is measurable by ELISA test, which is complemented by the HI assay, which is practical and specific for AI.

The data points were not aligned. Fig. 1 showed day 7 and day 14, however, Fig. 2 showed day 14 and day 21.

We appreciate your comments. As previously mentioned, both measurements of the local and systemic immune systems, due to the lag in the response speed of each system, were taken according to the moments with the greatest possibility of obtaining data for analysis.

There was no challenge experiment data, hence a more appropriate evaluation for the immunogenicity cannot be made.

We appreciate your comments. Due to the guidelines of the Mexican government, the testing of viral challenges with avian influenza is not authorized, except when presenting scientific support for the authorization of these experiments.

For Fig. 1, there was no significant difference between the total sIgA of treatment B to control at day 14. Likewise, the GMT of HI titers were below detection (<10) at day 14 for Fig. 2. 

We appreciate your comments. Both stimulations of the local and systemic immune response must be considered separately, they do not present a relationship in intensity and detection time, because both systems are considered complementary but separated in their functions. Therefore, the results can be considered complementary in the stimulation effect but not in the intensity of each response, since systemic stimuli are favored by parenteral applications.

The data at day 21 (Fig. 2) was different from their previous publication (Avian Dis 60: 784, 2016), i.e., the GMT HI titers after immunization (inactivated virus or recombinant HA) were below 20 at day 21 (Fig. 3 in that publication) but were at 40 in the current manuscript.

We appreciate your comments. In the present experiment, the experimental animals have different commercial origins, because these correspond to laying birds. Due to the presence of maternal immunity and different commercial origins, these parameters can be considered not similar to those obtained in previous reports.

In summary, this manuscript is incomplete. There was no new scientific knowledge gained. It should be rejected. Of note, these researchers were using a different species of single-cell algae (C. reinhardtii) previously. The reasons for the switch remain unknown. The reasons and possible differences should be fully explained, rather than just reporting similar data, a form of "self-plagiarism".

We appreciate your comments. The objective of this publication is to complement the information of a novel expression system based on a microalgae of the D.salina species. The previous report made above proposes the use of another model of microalgae expressing modified truncated proteins in chloroplast of a different species, so both experiments cannot be compared because both the expression model and the transformation system are completely different, as well as the characteristics of the recombinant proteins themselves are not comparable for various reasons (glycosylation, molecular weight, conformation, and quaternary structure)

Round 2

Reviewer 2 Report

The authors have addressed the comments raised

Author Response

Dear Reviewer, #2, thank you very much for the time you have spent reviewing our manuscript

Reviewer 3 Report

Dear editorial board

Authors replied to all comments and I feel that this research can be accepted now

Author Response

Dear Reviewer, #3, thank you very much for the time you have spent reviewing our manuscript.

Reviewer 4 Report

This superficial revision does not address the problems nor improve the overall scientific content of this manuscript. The claims listed in the abstract were not supported by any data provided. In addition, this revised manuscript creates more errors, e.g., AIV is the standard abbreviation for avian influenza virus, NOT AI. The expanded discussion, which is too long, should be moved to the background or introduction section. Some of the discussions had no relevance to the data provided, e.g., the discussion on lymphocyte aggregates was referenced from other studies; the authors provided no evidence to show that was the case in their immunized chickens.

This manuscript should only be reconsidered if the authors revise and shorten it into a brief report, or to provide relevant experimental data.

Author Response

Dear Reviewer, #4, thank you very much for the time you have spent reviewing our manuscript. Your comments are very valuable and helpful for revising our paper and guiding our research. We have studied those comments carefully and have made corrections, which we hope to meet with approval. The revised portion in the new version was included and is highlighted in yellow in the reviewed manuscript. The following is our point-by-point response to reviewers’ comments:

This superficial revision does not address the problems nor improve the overall scientific content of this manuscript. The claims listed in the abstract were not supported by any data provided. In addition, this revised manuscript creates more errors, e.g., AIV is the standard abbreviation for avian influenza virus, NOT AI. The expanded discussion, which is too long, should be moved to the background or introduction section. Some of the discussions had no relevance to the data provided, e.g., the discussion on lymphocyte aggregates was referenced from other studies; the authors provided no evidence to show that was the case in their immunized chickens.

This manuscript should only be reconsidered if the authors revise and shorten it into a brief report, or to provide relevant experimental data.

 We appreciate your comments and suggestions, which are accepted and included.
